


# Bayesian transdimensional inverse reconstruction of the $^{137}$Cs Fukushima-Daiichi release

Joffrey Dumont Le Brazidec[1,2], Marc Bocquet[2], Olivier Saunier[1], and Yelva Roustan[2]

[1]IRSN, PSE-SANTE, SESUC, BMCA, Fontenay-aux-Roses, France
[2]CEREA, École des Ponts and EDF R&D, Île-de-France, France

**Correspondence:** Joffrey Dumont Le Brazidec (joffrey.dumont@enpc.fr)

**Abstract.**

The accident at the Fukushima-Daiichi nuclear power plant yielded massive and rapidly varying atmospheric radionuclide releases. The assessment of these releases and of the corresponding uncertainties can be performed using inverse modelling methods that combine an atmospheric transport model with a set of observations and have proven to be very effective for this type of problem. In the case of Fukushima-Daiichi, a Bayesian inversion is particularly suitable because it allows errors to be modelled rigorously and a large amount of observations of different natures to be assimilated at the same time. More specifically, one of the major sources of uncertainty in the source assessment of the Fukushima-Daiichi releases stems from the temporal representation of the source. To obtain a well time-resolved estimate, we implement a MCMC sampling algorithm within a Bayesian framework, the Reversible-Jump MCMC, in order to retrieve the distributions of the magnitude of the Fukushima-Daiichi $^{137}$Cs source as well as its temporal discretisation. In addition, we develop Bayesian methods allowing to combine air concentration and deposition measurements, as well as to assess the spatio-temporal information of the air concentration observations in the definition of the observation error matrix.

These methods are applied to the reconstruction of the posterior distributions of the magnitude and temporal evolution of the $^{137}$Cs release. They yield a source estimate between 11 and 24 March, as well as an assessment of the uncertainties associated with the observations, the model and the source estimate. The total released reconstructed activity is estimated to be between 10 and 20 PBq, although it increases when taking into account the deposition measurements. Finally, the variable discretisation of the source term yields an almost hourly profile over certain intervals of high temporal variability, signaling identifiable portions of the source term.

## 1 Introduction

### 1.1 Fukushima-Daiichi nuclear accident

On 11 March 2011, an earthquake under the Pacific Ocean off the coast of Japan triggered an extremely destructive tsunami that hit the Japanese coastline about an hour later, killing 18,000 people. These events led to the automatic shutdown of four Japanese nuclear power plants. However the drowning of emergency power generators prevented the cooling system to function, making the shutdown impossible at the Fukushima-Daiichi nuclear power plant (NPP). This yielded a massive release





of radionuclides characterised by several episodes of varying intensity, which lasted for several weeks. The estimation of such a release is difficult and subject to significant uncertainties. Reconstructing the temporal evolution of the $^{137}$Cs source means to find a highly variable release rate running over several hundreds of hours.

## 1.2  Bayesian inverse modelling and sampling of a radionuclide source

Bayesian inverse problem methods have been shown to be effective in estimating radionuclide sources (Delle Monache et al., 30  2008; Tichý et al., 2016; Liu et al., 2017). Here, we briefly describe this Bayesian framework for source term reconstruction; a more complete description is available in Dumont Le Brazidec et al. (2021). The crucial elements defining the control variable vector of the source $\boldsymbol{x}$ are i) $\ln \boldsymbol{q}$, a vector whose components correspond to the logarithm of the release rates $q_i$ (constant releases on a time interval, such as a day or an hour), and ii) hyperparameters such as parameters of the observation error scale matrix $\mathbf{R}$ or the time windows of the release rates (more details in section 3).

The posterior probability density function (pdf) of $\boldsymbol{x}$ is given by Bayes' rule:

$$p(\boldsymbol{x}|\boldsymbol{y}) = \frac{p(\boldsymbol{y}|\boldsymbol{x})p(\boldsymbol{x})}{p(\boldsymbol{y})} \propto p(\boldsymbol{y}|\boldsymbol{x})p(\boldsymbol{x}) \tag{1}$$

with $\boldsymbol{y}$ the observation vector. The pdf $p(\boldsymbol{y}|\boldsymbol{x})$ is the likelihood, which quantifies the fit of the source vector $\boldsymbol{x}$ to the data. More precisely, the observations $\boldsymbol{y}$ are compared to a set of modelled concentrations: $\mathbf{H}\boldsymbol{q}$, i.e., the predictions, constructed using a numerical simulation of the radionuclide transport from the source. $\mathbf{H}$ is the observation operator, the matrix representing the 40  integration in time (i.e. resolvent) of the atmospheric transport model. Therefore, the predictions of the model are considered linear in $\boldsymbol{q}$. The likelihood is often chosen as a distribution parametrised with an observation error scale matrix $\mathbf{R}$. The pdf $p(\boldsymbol{x})$ represents the information available on the source before data assimilation.

Once the likelihood and the prior are defined, the posterior distribution can be estimated using a sampling algorithm. Sampling techniques include in particular the very popular Markov chain Monte Carlo (MCMC) methods that assess the posterior 45  pdf of the source, or marginal pdfs thereof. It has been applied by Delle Monache et al. (2008) to estimate the Algeciras incident source location, by Keats et al. (2007), or Yee et al. (2014) who evaluated Xenon-133 releases at Chalk River Laboratories. More recently, the technique was used to assess the uncertainties of the source reconstruction by Liu et al. (2017); Dumont Le Brazidec et al. (2020, 2021).

## 1.3  Transdimensional analysis and objectives of this study

The problem of finding a proper representation of $\boldsymbol{x}$, i.e. a discretisation of the source term, is the problem of finding the adequate step function for $\boldsymbol{q}$, i.e., the optimal number of steps, the optimal time length of each step and the corresponding release rates. This representation depends on the case under study: depending on the observations, more or less information is available to define a more or less resolved in time source term. In other words, depending on the data available to inform about the source term, the steps of the function supporting $\boldsymbol{q}$ could be small (a lot of available information allowing for a fine 55  representation) or large (few information available bringing a coarse representation), yielding an irregular discretisation in time of the source term. Note that the choice of the discretisation can be seen as a balancing issue due to the bias-variance trade-off





principle (Hastie et al., 2009): low complexity models are prone to high bias but low variance, while high complexity models are prone to low bias but high variance. The difficulty in the source term discretisation is that of selecting representations that are sufficiently rich, but not too complex to avoid over-fitting that leads to high variance error and too much computing time.

Radionuclides releases such as those of $^{137}$Cs from the Fukushima-Daiichi NPP are very significant over a long period of time (several weeks), and with a very high temporal variability. In such case, the choice of the discretisation is a crucial and challenging task.

In this paper we explore several ways to improve the source term assessment and its corresponding uncertainty. In the first instance, we address the issue of sampling in the case of massive and highly fluctuating releases of radionuclides to the

atmosphere. In the second instance, we propose several statistical models of the likelihood scale matrix depending on the available set of observations.

The outline of the paper is as follows. First, in section 2, previous works on the Fukushima-Daiichi NPP source term assessment are described as well as the observations dataset and the physical modelling of the problem.

Second, in section 3, we describe the theoretical aspects of this study and first focus on the concepts of transdimensionality

and model selection. The Reversible-Jump MCMC algorithm (RJ-MCMC) used to shape the release rates vector $q$ is presented in subsection 3.1. Then, other ways to improve the sampling quality and reduce uncertainties are explored through the addition of information in subsection 3.2. In particular, methods that combine deposition and concentrations observations and to exploit the temporal and spatial information of the observations in the definition of the likelihood scale matrix $\mathbf{R}$ are proposed.

Subsequently, these methods are evaluated in section 4. Specifically, once the statistical parametrisation of the problem is

chosen in subsection 4.1, the advantages of the RJ-MCMC are explored in subsection 4.2.1. The impact of using the deposition measurements and the use of the observation temporal and spatial information are described in subsections 4.2.2 and 4.2.3. We finally conclude on the contribution of each method.

This article is in line with the authors' earlier studies (Liu et al., 2017; Dumont Le Brazidec et al., 2020, 2021) and is inspired from the insightful work of Bodin and Sambridge (2009). In particular, the RJ-MCMC algorithm is applied by Liu et al. (2017)

on the Fukushima-Daiichi case with variable discretisations but with a fixed number of steps (i.e. grid cells). Here, it is here applied in a more ambitious transdimensional framework where the number of steps of the discretisation is modified through the progression of the RJ-MCMC, and using a significantly larger dataset.

## 2   Fukushima-Daiichi NPP $^{137}$Cs release and numerical modelling

### 2.1   Previous works

Since 2011, several approaches have been proposed to assess the Fukushima-Daiichi radionuclide release source terms. For instance, methods based on the simple comparison between observations and simulated predictions have been investigated by Chino et al. (2011); Katata et al. (2012); Mathieu et al. (2012); Terada et al. (2012); Hirao et al. (2013); Kobayashi et al. (2013); Katata et al. (2015); Nagai et al. (2017). Ambitious inverse problem methods applied to the source term have been developed (Stohl et al., 2012; Yumimoto et al., 2016; Li et al., 2019). In particular, Winiarek et al. (2012, 2014) used air concentration





and deposition measurements to assess the source term of $^{137}$Cs by rigorously estimating the errors of several datasets and the
prior; Saunier et al. (2013) evaluated the source term by inverse modelling with ambiant gamma dose measurements. Finally,
inverse problem methods based on a Bayesian formalism have been used. Liu et al. (2017) apply several methods including
MCMC algorithms to estimate the time evolution of the $^{137}$Cs release, accompanied by an objective estimate of the associated
uncertainty. Terada et al. (2020) refined the $^{137}$Cs source term with an optimisation method based on Bayesian inference which
used various measurement sources.

## 2.2 Observations dataset

The dataset used in this study consists of $^{137}$Cs air concentration and deposition measurements over the Japanese territory. The
vector of $^{137}$Cs air concentration measurements contain $14,248$ observations (Furuta et al., 2011; Yamada et al., 2013; Oura
et al., 2015; Nagakawa et al., 2015; Tanaka et al., 2013; Tsuruta et al., 2018; Takehisa et al., 2012) from $105$ stations, whose
locations are shown in Figure 1. This dataset is significantly larger than the one used by Liu et al. (2017). It allows us to better

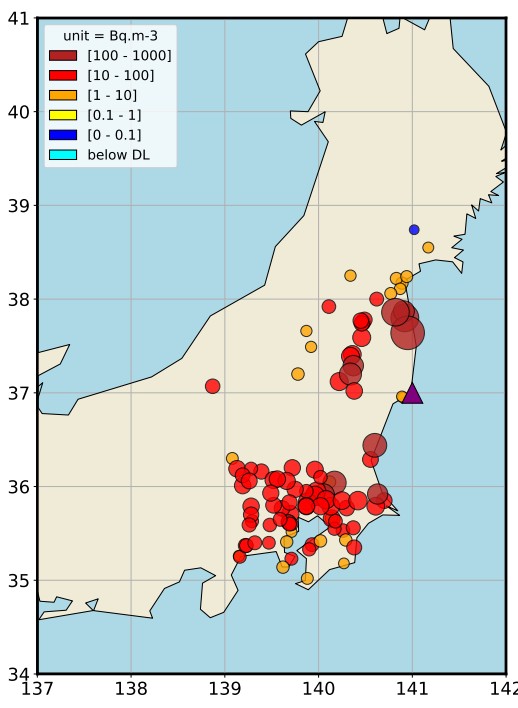

**Figure 1.** Maximum air concentrations of $^{137}$Cs in Bq.m$^{-3}$ measured in Japan in March 2011. The light blue points correspond to concentrations below the detection limit. The purple triangle corresponds to the Fukushima-Daiichi NPP.


evaluate the relevance of our methods and to accurately estimate the uncertainties of the problem. Moreover, the majority
of these observations are hourly, which allows for a fine sampling of the $^{137}$Cs source term. Additionally, 1507 deposition





measurements are used. This dataset is described and available in MEXT (2012). The observations are located within $300\,\mathrm{km}$ of the Fukushima-Daiichi NPP. However, measurements too close to the Fukushima-Daiichi plant (typically less than 5 times the spatial resolution of the meteorological fields described in section 1) are not used to take into account the short-range dilution effects of the transport model. These measurements represent the averages of each grid cell of a field of deposition observations in $\mathrm{Bq.m}^{-2}$ measured by aerial radiation monitoring.

### 2.3 Physical parametrisation

The atmospheric $^{137}$Cs plume dispersion is simulated with the Eulerian model ldX, already validated on the Chernobyl and Fukushima-Daiichi accidents (Quélo et al., 2007; Saunier et al., 2013), or again on the $^{106}$Ru releases in 2017 (Saunier et al., 2019; Dumont Le Brazidec et al., 2020, 2021). The meteorological data used are the three-hourly OPER OD ECMWF fields (see Table 1).

|  | ECMWF |
| --- | --- |
| Spatial resolution | $0.125° \times 0.125°$ |
| Time resolution | 3 hours |
| Vertical resolution | 11 terrain following layers (from 0 to 4400 m) |
| Vertical mixing | K-diffusion parametrised with Louis' closure (Louis, 1979) |
|  | and (Troen and Mahrt, 1986) in PBL unstable conditions |
| Horizontal mixing | constant horizontal eddy diffusion coefficient $K_h = 0\,\mathrm{m}^2\mathrm{s}^{-1}$ |
| Wet scavenging | $\lambda = \Lambda_0 p_0$ (below-cloud) with $\Lambda_0 = 5.10^{-5}\,\mathrm{h.(mm.s)}^{-1}$, |
|  | and $\lambda = \Lambda_1 p_0^{0.64}$ (in-cloud) with $\Lambda_1 = 5.10^{-4}\,\mathrm{h.(mm.s)}^{-1}$; |
|  | and $p_0$ is the rainfall intensity (Baklanov and Sørensen, 2001; QUEREL et al., 2021) |
| Dry deposition | constant deposition velocity $v_d = 2.10^{-3}\,\mathrm{m.s}^{-1}$ |

**Table 1.** Main setup parametrisations of the ldX $^{137}$Cs transport simulations. Spatial and time resolutions are those of the ECMWF meteorological fields and the transport model. The other entries correspond to parameters of the transport model only.

The radionuclides, and in particular $^{137}$Cs, were mainly deposited in the Fukushima Prefecture, north-west of the Fukushima-Daiichi NPP. The bulk of the releases occurred over a two-week period from 11 March on. Therefore, the model simulations start on 11 March 2011 at 00:00 UTC and end on 24 March 2011, corresponding to 312 hours during which radionuclides could have been released. The release is assumed to be spread over the first two vertical layers of the model (between 0 and 160m). The height of the release has a small impact because the modelled predictions are compared with observations sufficiently far away from the Fukushima-Daiichi NPP and the release mostly remains within the atmospheric boundary layer. Finally, it is assumed that all parameters describing the source aside from the reconstructed ones are known. This is for instance the case of the release height.



# 3 Methodological aspects of the inverse problem

## 3.1 Reversible-Jump MCMC

In this section, we describe the Bayesian Reversible-Jump MCMC algorithm used in the following to reconstruct the vector $\mathbf{q}$ discretised over a variable adaptive grid. A good description of the RJ-MCMC, which was introduced by Green (1995), can be

found in Hastie and Green (2012). The algorithm is also clearly explained and applied by Bodin and Sambridge (2009). The method was in particular used by Yee (2008) in the field of inverse modelling for the assessment of substance releases, and by Liu et al. (2017) to quantify Fukushima-Daiichi releases, but with a fixed number of variable size gridcells. A key asset of the RJ-MCMC technique is its ability to sample a highly variable release by balancing bias and variance errors.

The RJ-MCMC algorithm is a natural extension of the traditional Metropolis-Hastings (MH) algorithm but to transdimen-

sional discretisation grids. In a traditional MH, the distribution of the release is assessed through a succession of random walks (e.g., Dumont Le Brazidec et al., 2020). In particular, special random walks allow the pdf of the discretisation steps within the control vector $\boldsymbol{x}$ to be sampled.

Indeed, as described above, the function that describes the evolution of the release rate over time is a vector or step function where each step (i.e., a gridcell) defines a constant release rate over a time interval. When using the RJ-MCMC, these time

intervals are not fixed neither regularly spaced. An objective is hence to retrieve the best step-wise time partition of the source term, which defines the release rates.

To do this, we use the concept of boundary: a boundary is a time that separates two constant release rates, i.e., that separates two gridcells where the source term is defined. Let $\Lambda$ be the set of $N_b$ boundaries. Boundaries are 0, 312, and integers in

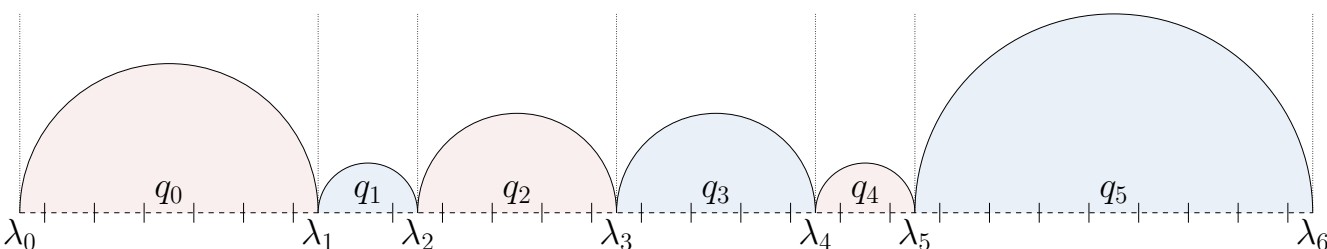

**Figure 2.** Example of a source term partition: $\Lambda = (\lambda_0, \ldots, \lambda_6)$ is the set of boundaries and $\boldsymbol{q} = (q_0, \ldots, q_5)$ is the vector of release rates. The release rate defined on the time interval $[\lambda_2, \lambda_3]$ is constant and equal to $q_2$.

$\{1, 2 \ldots, 311\}$ (0 and 312 are natural and fixed boundaries), i.e. the boundaries are a selection among the hours of the release.

Finding the probability distribution of the time steps associated to $\boldsymbol{q}$ is equivalent to finding the probability distribution of $\Lambda$. To retrieve the best discretisations of $\boldsymbol{q}$, we can therefore sample $\Lambda$.

The RJ-MCMC is basically equivalent to a MH algorithm except that some iterations of the algorithm are not intra-model random jumps (i.e., at constant $\Lambda$), but possibly inter-model random jumps (i.e., at non-constant $\Lambda$) or transdimensional (i.e., non-constant $\Lambda$ size).





Specifically, we use three types of random jumps that govern the positions and number of boundaries: birth, death, and move. For these three processes, we draw on the work of Green (1995), Hastie and Green (2012), Liu et al. (2017), and importantly Bodin and Sambridge (2009). The birth process is the creation of a new boundary not present in $\{1, 2, \ldots, 311\}$ yet; the death process is the removal of an existing boundary in $\Lambda$, and the move process is the displacement of an existing boundary in $\Lambda$. These processes must respect the detailed balance criterion in order to ensure the convergence of the Markov chain to the target 150  distribution (Robert et al., 2018).

### 3.2   Handling observations and their errors

#### 3.2.1   Measurements fusion

In this subsection, we propose a method to factor both concentration and deposition measurements into a Bayesian sampling by simply taking into account their disparity in the observation error matrix $\mathbf{R}$. First of all, $\mathbf{R}$ is assumed diagonal, hence a 155  collection of scale parameters. Specifically, two scale parameters are used when modelling $\mathbf{R}$: $r_c$ and $r_d$, associated to the air concentration and deposition measurements, respectively. We can write:

$$\mathbf{R} = \mathrm{diag}(r_c, \ldots, r_c, r_d, \ldots, r_d), \tag{2}$$

where $\mathrm{diag}(\{r_i\}_i)$ is the diagonal matrix of diagonal entries $\{r_i\}_i$. Note that in all subsequent applications, we use the sorting algorithm described by Dumont Le Brazidec et al. (2021) which is applied on both the air concentration and deposition 160  measurements. This algorithm assigns a different scale parameter $r_{\mathrm{np}}$ to certain non-pertinent observations to prevent them to reduce artificially $r_c$ and $r_d$. With this definition of $\mathbf{R}$, the scale variables $r_c, r_d, r_{\mathrm{np}}$ are included in the source vector $\boldsymbol{x}$ and therefore sampled using a MCMC.

#### 3.2.2   Spatio-temporal observation error scale matrix modelling

We propose in this subsection to model $\mathbf{R}$ using the spatio-temporal information of the concentration observations. We assume 165  that the error scale parameter corresponding to an observation-prediction pair depends mainly on the error in the model predictions. It is also proposed to consider a prediction, i.e., an estimation of the amount of radionuclides present at a definite time and place, as a point in a radionuclide plume at a specific time. The prediction modelling error varies according to its spatio-temporal position: spatially, if two points of interest are distant by $\Delta_x$, then the difference of errors in their attached model prediction is empirically assumed to be proportional to $\Delta_x$. Temporally, let us consider two points in a plume with coordinates 170  $(x_1, x_2)$ at times $t$ and $t + \Delta_t$, respectively, where $x_1$ and $x_2$ are the longitude and latitude. These two points are moving with the plume, and at time $t$, they are spatially distant by $v \times \Delta_t$ where $v$ is a reference wind speed, representing the average wind over the accidental period. Therefore the difference of the modelling errors of these points is estimated to be proportional to $\Delta_x = v \times \Delta_t$. In that respect, to each concentration observation $y_i$ can be associated three coordinates:

$$\left( x_{1,i}, x_{2,i}, v \cdot \left( t_i + \frac{\Delta_{t,i}}{2} \right) \right) \tag{3}$$





where $x_{1,i}$, $x_{2,i}$ are the coordinates in km of the observation, $v$ is set equal to $12\,\mathrm{km.h}^{-1}$, $t_i$ and $\Delta_{t,i}$ are the starting time and the duration of the observation in hours. This average wind speed value has not been extensively researched and is only used to estimate the potential of the method. A clustering algorithm such as the k-means algorithm can be applied to partition these three-coordinates observation characteristics in several groups. Then, a scale parameter $r_{c,i}$ can be assigned to each group. If the algorithm for sorting pertinent and non-pertinent observations is applied, this yields

$$\mathbf{R} = \mathrm{diag}(r_{c,i}, r_{\mathrm{np}}, r_{c,j}, \ldots, r_d, r_{\mathrm{np}}) \tag{4}$$

with $r_{c,i}, r_d, r_{\mathrm{np}}$ the scale parameters associated to the spatial or spatio-temporal clusters of air concentration observations, the deposition measurements, and the non-pertinent observations, respectively. The case where only the spatial qualification of the observation is used instead of both the spatial and temporal qualification is also addressed in this study.

## 4 Application to the Fukushima-Daiichi NPP $^{137}$Cs release

In this section we apply the previously introduced methods to the reconstruction of the Fukushima-Daiichi $^{137}$Cs source term between 11 and 24 March. But we first define the key statistical assumptions and parameters of the inversion setup.

### 4.1 Statistical parametrisation

The prior distributions determining the posterior distribution and the transition probabilities of the RJ-MCMC algorithm are defined in the next two sections.

#### 4.1.1 Definition of the prior density functions

In this section, we specify the prior distributions of the source vector components: release rates and hyperparameters. The logarithm of the source rates are assumed to follow a folded Gaussian prior distribution, so as to constrain at the same time the number of boundaries and the magnitude of the release rates. In the general case with $\mathrm{N}_{\mathrm{imp}} = 312$, the scale matrix $\mathbf{B}$ of the folded Gaussian prior is defined as a $312 \times 312$ diagonal matrix. The entry $b_k$ of $\mathbf{B}$ is associated to the $k$-th hour in

$\{0, 1, \ldots, 311\}$, hours where we consider the release of $^{137}$Cs possible. This yields:

$$p(\ln \boldsymbol{q} | \mathrm{N}_{\mathrm{imp}}) = \prod_{k=0}^{\mathrm{N}_{\mathrm{imp}}-1} \sqrt{\frac{2}{\pi b_k}} \left( e^{-\frac{(\ln q_k - \ln q_{\mathrm{b}})^2}{2 b_k}} \right) \tag{5}$$

where the folding position is defined as equal to the location term $\ln q_{\mathrm{b}}$. We divide the 312 time intervals into two groups: the first "unconstrained" group consists of the time intervals during which we have noticed that a radionuclide emission does not trigger any noticeable observation. In other words, it is a time interval during which we have no information on a potential re-

lease. The unconstrained time intervals are the first 17 hours of 11 March (hours in $\{0, 1, \ldots, 16\}$), the days of 16 and 17 March (hours in $\{116, 117, \ldots, 165\}$), and the last 18 hours of 23 March (hours in $\{294, 295, \ldots, 311\}$). The second group consists of the constrained time intervals. Two hyperparameters $b_{\mathrm{c}}$, for the constrained time intervals, and $b_{\mathrm{nc}}$, for the unconstrained ones,





are used to describe the matrix $\mathbf{B}$. If, instead of two, a single hyperparameter $b$ is used for all release rates, the unconstrained
time intervals are neither regularised by the observations nor by the prior. Indeed, to account for for high release rates, samples
of the distribution of $b$ reach significant values. Low magnitude release rates are then very little constrained by the corresponding prior. Release rates which are neither constrained by the observations nor by the priors can, in turn, reach very large values.
This phenomenon is analysed by Liu et al. (2017). The mean of the folded Gaussian prior is chosen as $\ln q_{\mathrm{b}} = 0$ in order to
regularise the unconstrained intervals. It is important to note that a release rate $\ln q_i$ sampled by the RJ-MCMC is an aggregate
of these hourly release rates, each of them associated with either $b_c$ or $b_{nc}$. We hence define the scale of the prior associated
with the release rate $\ln q_i$ defined between times $t_i$ and $t_i + k_i \Delta_t$ as

$$b_i = \frac{\sum_{j=t_i}^{t_i + k_i \Delta_t} \mathbf{B}_{j,j}}{k_i \Delta_t} = \frac{n_{c,i} b_{\mathrm{c}} + (k_i \Delta_t - n_{c,i}) b_{\mathrm{nc}}}{k_i \Delta_t} = w_{\mathrm{c},i} b_{\mathrm{c}} + w_{\mathrm{nc},i} b_{\mathrm{nc}} \tag{6}$$

with $\mathbf{B}_{j,j}$ the $j$-th diagonal coefficient of $\mathbf{B}$, $n_{c,i}$ the portion of constrained time intervals in $k_i \Delta_t$, and with $w_{\mathrm{c},i}$ and $w_{\mathrm{nc},i}$
the weighted numbers of constrained and unconstrained hourly release rates included in the release rate $\ln q_i$. This definition is
reflected in the prior cost:

$$\mathbf{J}_{\mathrm{prior}, \ln \boldsymbol{q}} = \sum_{i=0}^{\mathrm{N}_{\mathrm{imp}}-1} \left\{ \frac{1}{2} \ln \left( \frac{\pi b_i}{2} \right) + \frac{(\ln q_i)^2}{2 b_i} \right\} \tag{7}$$

$$= \sum_{i=0}^{\mathrm{N}_{\mathrm{imp}}-1} \left\{ \frac{1}{2} \ln \left( \frac{\pi (w_{\mathrm{c},i} b_{\mathrm{c}} + w_{\mathrm{nc},i} b_{\mathrm{nc}})}{2} \right) + \frac{(\ln q_i)^2}{2 (w_{\mathrm{c},i} b_{\mathrm{c}} + w_{\mathrm{nc},i} b_{\mathrm{nc}})} \right\} \tag{8}$$

where $\mathrm{N}_{\mathrm{imp}} = \mathrm{N}_{\mathrm{b}} - 1$ is the number of time steps ($\mathrm{N}_{\mathrm{b}}$ being the number of boundaries) considered at a certain iteration by the
RJ-MCMC and which characterises the grid on which $\ln \boldsymbol{q}$ is defined. The prior scale parameter $b_c$ is included in the source
vector $\boldsymbol{x}$ and sampled with the rest of the variables. Such a prior on the release rate logarithms also has a constraining effect on
the model's complexity (Occam's razor principle) through the normalisation constant of the prior pdf.

An exponential prior distribution is used for the boundaries $\Lambda = (\lambda_1, \ldots, \lambda_k)$:

$$p(\lambda_1, \ldots, \lambda_k) = \begin{cases} \frac{e^{-k}}{\sum_{i=1}^{\mathrm{N}_{\mathrm{b,max}}} \frac{\mathrm{N}_{\mathrm{b,max}}!}{i!(\mathrm{N}_{\mathrm{b,max}}-i)!} e^{-i}}, & \text{if } k \in \{1, 2, \ldots, \mathrm{N}_{\mathrm{b,max}}\}; \\ 0 & \text{otherwise,} \end{cases} \tag{9}$$

with $\mathrm{N}_{\mathrm{b,max}} = 311$. This prior has the effect of penalising models that are too complex.

Furthermore, because no a priori information is available, the prior distributions on the coefficients of the observation error
scale matrix are assumed to be uniform and the prior on the regularisation scale terms are assumed to follow Jeffreys' prior
distribution (Jeffreys, 1946; Liu et al., 2017). The lower and upper bounds of the uniform prior on the coefficients of the
observation error matrix are set as $0$ and a large value, not necessarily realistic, as in Dumont Le Brazidec et al. (2021).

### 4.1.2 Parameters of the MCMC algorithm

In the Markov chain, the variables describing the source $\boldsymbol{x}$ are initialised randomly. The intra-model (i.e., with a fixed boundary
partition) transitions defining the stochastic walks are set independently for each variable. Each transition probability is defined





as a folded-normal distribution, following Dumont Le Brazidec et al. (2020). Locations of the transition probabilities are the values of the variables at the current step. Variances of the related normal distribution are initialised at $\sigma_{\ln q} = 0.3$, $\sigma_r = 0.01$, the values being chosen by means of experiments. However, the value of $\sigma_r$ is adaptive and is updated every $1000$ iterations according to the predictions values. Furthermore, the value of $\sigma_r$ is not uniform across groups of observations: there are as many variances as observation clusters.

The algorithm runs initially as a classic MH for $10^4$ iterations (only intra-model walks are allowed at this stage). Indeed, allowing transdimensional walks at the beginning of the run sometimes cause Markov chains to get stuck in local minima. In total, $2 \times 10^6$ iterations are used which is a large number ensuring the algorithm convergence (i.e. a sufficient sampling of the posterior distribution of $\boldsymbol{x}$) and corresponding to approximately a day of calculation for a 12-core computer. The burn-in is set to $10^6$ iterations.

The inter-model transdimensional walks (with changing boundaries) are described in appendix A since it is technical. For this transdimensional case, $u_{\ln q}$ is the same for birth and death processes and is equal to $0.3$. There is no variance parameter for the move walk. The probability of proposing an intra-model transition is $1/2$, of proposing a birth transition is $1/6$, of proposing a death transition is $1/6$ and a move transition is $1/6$.

## 4.2 Results

To evaluate the quality of our reconstructed source term as well as the impact of the techniques proposed in section 3, the distributions of the $^{137}$Cs source are sampled and compared under several settings:

- subsection 4.2.1 provides an activity concentration-based reconstruction of the $^{137}$Cs source with a log-Cauchy likelihood and a threshold, to ensure it is defined for zero observations or predictions, of $0.5\,\mathrm{Bq.m}^{-3}$ (Dumont Le Brazidec et al., 2021), the application of the observation sorting algorithm, and a spatial representation of $\mathbf{R}$ with 10 parameters. The vector

$$\boldsymbol{x} = \left(\ln \boldsymbol{q}, r_{c,1}, \ldots, r_{c,9}, r_{\mathrm{np}}, b_c, (\lambda_1, \ldots, \lambda_{\mathrm{N}_{\mathrm{imp}}-1})\right) \tag{10}$$

  is sampled. A log-Cauchy likelihood is chosen because the associated reconstruction produces the best model-to-data comparisons among diverse choices of likelihoods and representations of $\mathbf{R}$. The sampled source is compared to the source terms of Saunier et al. (2013) and Terada et al. (2020) and all variable marginal distributions are described;

- in subsection 4.2.2, we are interested in quantifying the impact of the addition of deposition measurements in the sampling with the method presented in subsection 3.2.1. The source term is sampled in two configurations: with or without the addition of deposition measurements. That is, the vector

$$\boldsymbol{x} = \left(\ln \boldsymbol{q}, r_{c,1}, \ldots, r_{c,9}, r_{\mathrm{np}}, r_d, b_c, (\lambda_1, \ldots, \lambda_{\mathrm{N}_{\mathrm{imp}}-1})\right). \tag{11}$$

  is sampled with or without $r_d$. A threshold of $0.5\,\mathrm{Bq.m}^{-2}$ is chosen for the deposition measurements.;





– finally, we draw a selection of violin plots in subsection 4.2.3 to assess the impact of the distribution of the observations in space and time clusters using the approach introduced in subsection 3.2.2. More precisely, the fitness to the observations of the reconstructed source is evaluated given the type of clustering (spatial or spatio-temporal) and the number of clusters.

### 4.2.1 Best reconstruction and comparison to other source terms

In this section we look for the most appropriate reconstruction of the source term from several choices of likelihoods and representations of $\mathbf{R}$ according to a model-to-data comparison, i.e., FAC scores, which represent the proportion of measurements for which the observed and modelled values agree. The chosen likelihood is a log-Cauchy distribution with a threshold of $0.5\,\mathrm{Bq.m^{-3}}$. Only air concentration measurements are used and they are spatially clustered in 9 groups using the k-means method.

In figure 3 is shown the temporal evolution of the median release rate with its associated standard deviation, i.e. the evolution of the hourly release rates samples that are obtained from the $\mathrm{N_{imp}}$ RJ-MCMC samples. This temporal evolution is compared to the source term from Saunier et al. (2013) which was estimated from gamma dose rate measurements, but from the same meteorological data and transport model, and to the more recent source term from Terada et al. (2020).

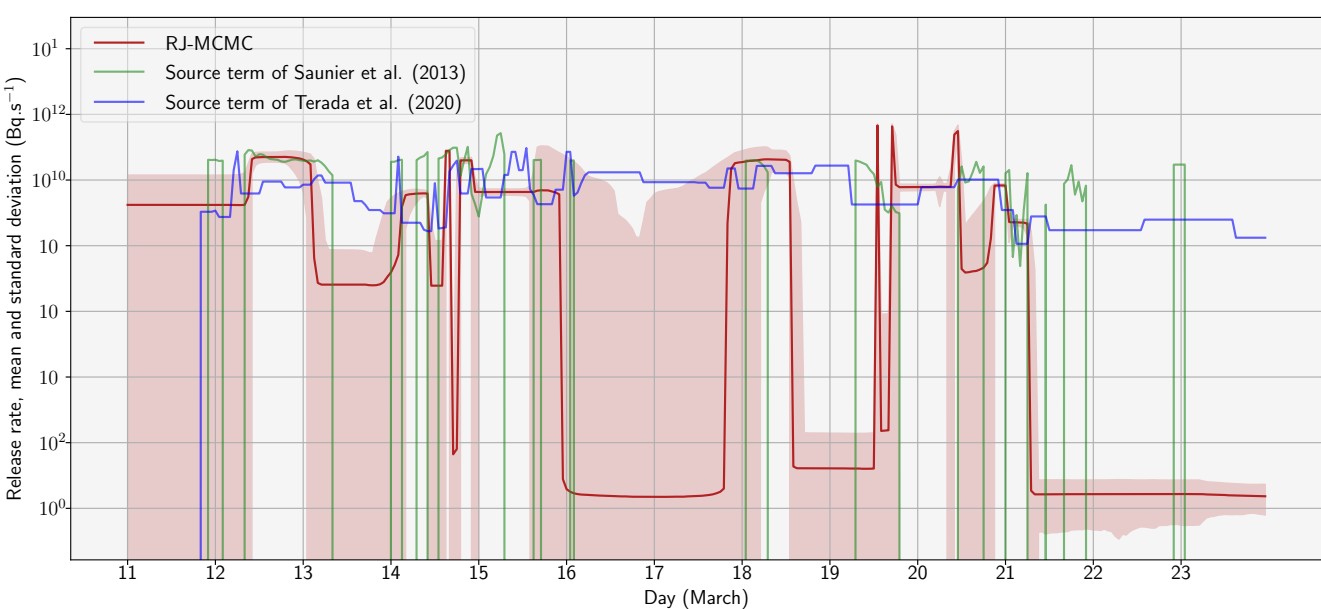

**Figure 3.** Evolution of the release rate in Bq.s$^{-1}$ describing the $^{137}$Cs source (UTC) sampled with the RJ-MCMC algorithm. The red line corresponds to the sampled median release rate. The light red area corresponds to the area between $\mu_{\ln q}$ the median and $\sigma_{\ln q}$ the standard deviation of the reconstructed hourly release rates: $[\mu_{\ln q} - \sigma_{\ln q}, \mu_{\ln q} + \sigma_{\ln q}]$. Our source term is compared to the source terms of Saunier et al. (2013) and Terada et al. (2020).





Firstly, we observe that a good match is obtained between the median release and the source term of Saunier et al. (2013). On the other hand, on the constrained time intervals, our source term and that of Terada et al. (2020) are similar in magnitude, as the differences between releases under $10^9\,\mathrm{Bq.s^{-1}}$ can be neglected. Secondly, release rates are subject to large variations. Indeed, there are periods of important temporal variability of the release rate (e.g., between 19 and 21 March), and of low temporal variability (e.g., between 11 and 14 March). This justifies the use of the RJ-MCMC transdimensional algorithm,

as it allows to reconstruct certain time intervals of the release with a greater accuracy and others with a lesser complexity. Thirdly, the temporal evolution of the release can be explored and several prominent peaks can be observed with a magnitude approaching or surpassing $10^{11}\,\mathrm{Bq.s^{-1}}$: between 12 and 14 March, between 14 and 16 March, on 18 March, and between 19 and 21 March.

Panel a of figure 4 describes the histogram of relative differences in $\log_{10}$ between predictions (medians of the predictions

computed using the samples are used) and bigger than $0.5\,\mathrm{Bq.m^{-3}}$ air concentration observations. A value of -1 therefore corresponds to an observation 10 times larger than a prediction. A good fit between observations and predictions can be observed although the observations are globally underestimated by the predictions. The FAC 2 and FAC 5 scores of the combined air concentration and deposition predictions are $0.321$ and $0.649$, respectively.

Panel b of figure 4 shows the pdf of the total release in Bq. It can be seen from this density that most of the total release

distribution mass is sampled between 10 and 20 PBq and peaks at 14 PBq. This is consistent with previous studies which estimate the release to be between 5 and 30 PBq (Chino et al., 2011; Terada et al., 2012; Winiarek et al., 2012, 2014; Saunier et al., 2013; Katata et al., 2015; IAEA, 2015; Yumimoto et al., 2016; Liu et al., 2017; Terada et al., 2020). However, note that these works attempt to estimate potential releases beyond 24 March.

Panel c of figure 4 plots the pdf of the number of boundaries, i.e. a measure of model complexity, sampled by the algorithm.

Here, the minimum of the distribution is 25 boundaries and the maximum is 45 with a peak at 35 boundaries. Recall that the maximum number of boundaries is 313.

Finally, panel d of figure 4 shows the average number of sampled boundaries at each time and the associated standard deviation. The y-axis corresponds to the number of boundaries counted: for $t$ a given hour, we draw:

$$\mathrm{E}[\mathrm{Card}\{\lambda_i \in [t-5\ldots t+5]|\lambda_i \in (\lambda_0,\ldots,\lambda_{\mathrm{N_{imp}}})\}] \tag{12}$$

i.e. the average over all samples of the number of boundaries counted around $t$. In essence, the curve shows the number of boundaries necessary to model the release at a certain time, i.e., the variability of the release according to the algorithm. We observe that there is a large variability between 14 and 15 March, and between 19 and 21 March. This correlates very well with the observable results of Figure 3. This observation confirms once again the relevance of using the RJ-MCMC algorithm: periods of high variability can be sampled finely.

### 4.2.2    Impact of using deposition measurements

In this section, we study the impact of adding $1{,}507$ deposition measurements to the $14{,}248$ air concentration measurements. A log-Cauchy distribution with a threshold of $0.5\,\mathrm{Bq.m^{-3}}$ for the air concentration measurements and $0.5\,\mathrm{Bq.m^{-2}}$ for the deposit



**Figure 4.** Densities or averages of variables describing the $^{137}$Cs source reconstructed with the RJ-MCMC technique.

(a) Density of relative differences in $\log_{10}$ between observations and predictions. (b) Density of the Total Retrieved Released Activity (TRRA) of $^{137}$Cs during the Fukushima-Daiichi accident in Bq; outliers are removed. (c) Histogram of the number of boundaries. (d) Mean (± standard deviation) of the number of boundaries as a function of time (UTC).





measurements is employed. Air concentration measurements are clustered with a spatial k-means algorithm in 9 groups in the same way as described in section 4.2.1. To quantify the impact of the added measurements we sample the following vector

with a RJ-MCMC algorithm in two cases, with or without deposition measurements, i.e., with or without $r_d$:

$$\boldsymbol{x} = \left(\ln \boldsymbol{q}, r_{c,1}, \dots, r_{c,9}, r_{\text{np}}, r_d, b_c, (\lambda_1, \dots, \lambda_{\text{N}_{\text{imp}}-1})\right). \tag{13}$$

In the case without deposition measurements, we use the same samples as in subsection 4.2.1. A posteriori marginals of these control variables are displayed in Figure 5. In figure 5.a, the median of the two samplings are similar, except for the release rates between 11 and 14 March. Indeed, when using air concentration and deposition measurements, the release rate peaks

during 12 March instead of 13 March when using air concentrations alone.

In panel b of figure 5, the main mode of the TRRA distribution retrieved using the air concentration and deposition measurements is $5\,\text{PBq}$ larger than the TRRA reconstructed with the air concentration measurements alone. Besides, the reconstructed density with both types of measurements has a large variance and leads to the possibility of a larger release (up to $75\,\text{PBq}$). The density of the number of boundaries with both types of measurements is shifted to higher values compared to the density with

air concentrations alone, which indicates a reconstructed release of greater complexity (panel c of figure 5). This is consistent with the fact that the second release is reconstructed with a larger number of observations.

In panel d of figure 5, the number of boundaries was estimated to be half as large on 14 March when deposition measurements were used, but was estimated to be larger on March 15. These differences are rather surprising insofar as there is no obvious disparity between the medians of the release rates for the two reconstructions in panel a of figure 5 on these days.

In table 2, we evaluate the FAC 2, FAC 5, FAC 10 scores for both cases. As anticipated, it shows that there is a significant dif-

| Sampling | FAC 2 | FAC 5 | FAC 10 |
|---|---|---|---|
| With air concentrations only | | | |
| air concentration | 0.32 | 0.68 | 0.85 |
| deposition | 0.31 | 0.58 | 0.74 |
| With air concentration and deposition | | | |
| air concentration | 0.31 | 0.67 | 0.85 |
| deposition | 0.48 | 0.76 | 0.85 |

**Table 2.** Comparison of observed and simulated measurements from reconstructed predictions with air concentration measurements only, or air concentration and deposition measurements. The FAC 2, FAC 5, FAC 10 are the proportion of measurements for which the observed and modelled values agree by a factor of 2, 5, or 10, respectively. The table reads as follows: e.g., the predictions based on air concentrations only, obtain a FAC 5 score of 0.58 with respect to the deposition measurements.


ference in the reproduction of the deposition measurements when they are added to the observation dataset. On the other hand, adding the deposition measurements does not impact the quality of the reconstruction of the air concentration measurements.



**Figure 5.** The panels correspond to probability densities or averages of variables describing the $^{137}$Cs source reconstructed with the RJ-MCMC technique using air concentration and deposition measurements. (a) Medians and standard deviations of hourly release rates: the red and blue curves correspond to the sampled median release rates. The red and light blue areas correspond to the areas between $\mu_{\ln q}$ the median and $\sigma_{\ln q}$ the standard deviation of the sampled hourly release rates for each reconstruction: $[\mu_{\ln q} - \sigma_{\ln q}, \mu_{\ln q} + \sigma_{\ln q}]$. (b) Probability density of the total release of $^{137}$Cs during the Fukushima-Daiichi accident in Bq. (c) Histogram of the number of boundaries. (d) Mean ($\pm$ standard deviation) of the number of boundaries according to the hour considered.





### 4.2.3 Impact of the observation error matrix $\mathbf{R}$ representation

We investigate in this section the spatial and spatio-temporal representation of the $\mathbf{R}$ likelihood scale matrix. In figure 6 are
drawn violin plots of the FAC 2 and FAC 5 scores for air concentration measurements against the number of scale parameters
chosen to describe $\mathbf{R}$, i.e., the number of observations clusters. Violin plots are created out of the best scores of 44 RJ-MCMC
samplings with diverse likelihoods chosen after Dumont Le Brazidec et al. (2021) in order to make the conclusions less
dependent on statistical modelling. The chosen numbers of spatial clusters (1, 2, 9) and spatio-temporal clusters (3, 7) of the
violins are a selection for the sake of clarity. It can be observed for both FAC 2 and FAC 5 air concentration scores that the

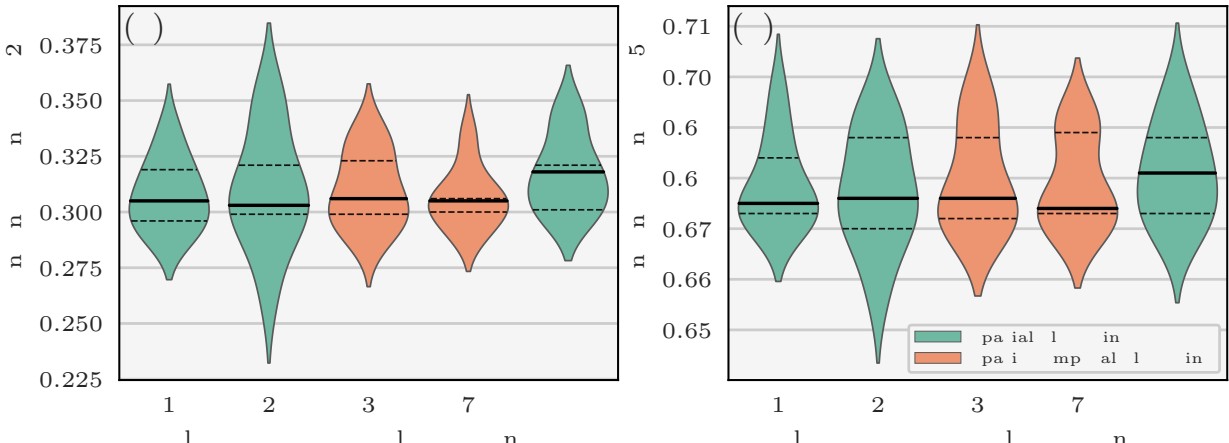

**Figure 6.** Violin plots of the air concentration FAC 2, FAC 5 scores against the observation error scale parameters spatio-temporal groups
number. Points composing the violin plots consist of the best FAC 2, FAC 5 scores of 44 samplings in diverse configurations of likelihoods
and thresholds (log-normal, log-Laplace, and log-Cauchy) chosen according to Dumont Le Brazidec et al. (2021). The horizontal segments
within the violins represent from bottom to top the first, second and third quartiles, respectively. The chosen numbers of spatial and spatio-
temporal clusters are only a selection for the sake of clarity.

spatio-temporal labelling of the observations does not increase the quality of the scores, while the number of spatial clusters
does. We can observe for example a difference of 0.02 between the second quartile of the FAC 2 score for samplings with 2
spatial groups and the second quartile of the FAC 2 score for samplings with 9 spatial groups. One hypothesis is that the error
is mainly related to the distance to the source and remains relatively homogeneous in time. However, the use of an accurate
estimate of the effective wind speed as a function of time and depending on the position of the observations is necessary to
draw conclusions; such a study is outside the scope of this paper.





## 5 Summary and conclusions

In this paper, we investigated a transdimensional sampling method to reconstruct highly fluctuating radionuclide atmospheric sources and applied it to assess the $^{137}$Cs Fukushima-Daiichi release. Furthermore we proposed two more methods to add information in the model in order to reduce uncertainties attached to such a complex release.

Firstly, a RJ-MCMC algorithm was defined. This MH extension to transdimensional meshes allows to sample the both source term and its underlying adaptive discretisation in time. In the case of complex releases, an adaptive grid allows to reconstruct the source term with higher accuracy and to reduce the corresponding uncertainties.

    Secondly, we have focused on ways to add information by adapting the representation of the observation error matrix $\mathbf{R}$. We proposed to assimilate deposition measurements or spatial and temporal information on the air concentration observations by

increasing the complexity of $\mathbf{R}$.

    Subsequently, the distributions of the variables defining the source of $^{137}$Cs were fully sampled. This enabled the estimation of the uncertainties associated with these variables, as well as the evaluation and demonstration the merits of the methods.

    Firstly, we observed that the predictions reproduce well the air concentration and deposition observations. When using air concentrations only, the FAC 2 and FAC 5 scores corresponding to air concentration measurements are $0.32$ and $0.68$,

respectively. The main part of the best total release distribution mass is estimated to be between $10$ and $20\,\mathrm{PBq}$ and peaks at $14\,\mathrm{PBq}$ which is in accord with previous estimations.

    Secondly, the periods of high variability of $^{137}$Cs releases have been reconstructed with accuracy by the RJ-MCMC algorithm. The transdimensional method also allowed the periods of low temporal variability to be sampled more coarsely, thus avoiding variance errors and saving computing time, which, in particular, confirms the conclusions of Liu et al. (2017). The

priors we chose allowed for an efficient regularisation of the release rates and their adaptive mesh.

    Finally, the use of combined air concentration and deposition measurements had an impact on the reconstruction of the TRRA distribution, the mean of which increased by $10\,\mathrm{PBq}$. The spatial clustering methods proved to increase the quality of the predictions by improving model-to-data comparison. However, the addition of temporal information on the concentration observations did not improve the prediction quality.

We recommend the use of the RJ-MCMC for long release source reconstruction. It allows to sample highly variable source terms with great accuracy by solving the bias-variance tradeoff. We also recommend the use of advanced modelling of $\mathbf{R}$ taking in account the deposition measurements and the spatial information of the air concentration observations which have proven to be valuable in reducing the observation-prediction discrepancies.

## Appendix A: RJ-MCMC mathematical details

We describe here the mathematical details of the three inter-model walks: move, birth and death. The move process is intra-dimensional. Provided that the move is a symmetric motion, the associated $g$-transition distribution is symmetric, i.e.:

$$\frac{g(\boldsymbol{x}_i|\boldsymbol{x}_j)}{g(\boldsymbol{x}_j|\boldsymbol{x}_i)} = 1. \tag{A1}$$





The birth process is the creation of a new boundary, a walk from $\boldsymbol{x}_i$ a vector composed of $n$ boundaries to $\boldsymbol{x}_j$ a vector composed of $n+1$ boundaries is proposed. In other words, by generating a new boundary, one release rate is destroyed and two new release rates emerge from this destruction. We therefore assign two new release rates $\ln q'_{k-1} = \ln q_{k-1} - u_{\ln q}$ and $\ln q'_k = \ln q_{k-1} + u_{\ln q}$ with $u_{\ln q}$ a Gaussian noise. The transition probability is defined as follows (Bodin and Sambridge, 2009):

$$g(\Lambda_i, \ln \boldsymbol{q}_i | \Lambda_j, \ln \boldsymbol{q}_j) = g(\Lambda_i | \Lambda_j, \ln \boldsymbol{q}_j) \, g(\ln \boldsymbol{q}_i | \Lambda_i, \Lambda_j, \ln \boldsymbol{q}_j) = g(\Lambda_i | \Lambda_j) \, g(\ln \boldsymbol{q}_i | \ln \boldsymbol{q}_j, \Lambda_i). \tag{A2}$$

The probability of a birth at a certain position among $\mathrm{N_{b,max}} - \mathrm{N_{imp}} + 1$ positions not occupied by boundaries is

$$g(\Lambda_i | \Lambda_j) = \frac{1}{\mathrm{N_{b,max}} - \mathrm{N_{imp}} + 1}. \tag{A3}$$

Furthermore, the probability of generating new release rates is Gaussian:

$$g(\ln \boldsymbol{q}_i | \Lambda_i, \ln \boldsymbol{q}_j) = g(\ln q'_{k-1}, \ln q'_k | \ln q_{k-1}, u_{\ln q}) = \frac{1}{\sqrt{2\pi}\sigma} e^{-\frac{u_{\ln q}^2}{2\sigma^2}}, \tag{A4}$$

where the new boundary is here generated at hour $k-1$.

The probability of death of one among the $\mathrm{N_{imp}} - 1$ boundaries is

$$g(\Lambda_j | \Lambda_i) = \frac{1}{\mathrm{N_{imp}} - 1}, \tag{A5}$$

which is a uniform choice among $\mathrm{N_{imp}} - 1$ possibilities. On the other hand, the probability $g(\ln \boldsymbol{q}_j | \Lambda_j, \Lambda_i, \ln \boldsymbol{q}_i)$ of destroying the release associated to this position is 1.

*Data availability.* Each dataset used in the manuscript is available via a DOI on a compliant repository or directly in the cited papers. More precisely, the air concentration observation datasets used in the inversions are available in (Oura et al., 2015) (at http://www.radiochem.org/en/j-online152.html) and in (Tsuruta et al., 2018; Tanaka et al., 2013; Takehisa et al., 2012; Yamada et al., 2013). In addition, the deposition observation dataset used in the inversions is available via 10.5281/zenodo.7016490.

*Author contributions.* Joffrey Dumont Le Brazidec: Software, Methodology, Conceptualisation, Investigation, Writing - Original Draft, Visualisation. Marc Bocquet: Methodology, Software, Conceptualisation, Writing - Review and Editing, Visualisation, Supervision. Olivier Saunier: Resources, Methodology, Conceptualisation, Writing - Review and Editing, Visualisation, Supervision. Yelva Roustan: Methodology, Software, Writing - Review and Editing.

*Competing interests.* The authors declare that they have no conflict of interest.





*Acknowledgements.* The authors would like to thank the European Centre for Medium-Range Weather Forecasts (ECMWF). They would also like to thank Sophie Ricci, Lionel Soulhac, Paola Cinnella, Didier Lucor, Yann Richet and Anne Mathieu for their suggestions and advice on this work. CEREA is a member of Institut Pierre-Simon Laplace (IPSL).





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
