# Peer review of "Bayesian transdimensional inverse reconstruction of the $^{137}\mathrm{Cs}$ Fukushima-Daiichi release"

_Geoscientific Model Development, 2022_

## Referee Comment (RC1)

**Review of gmd-2022-168:** *Bayesian transdimensional inverse reconstruction of the $^{137}Cs$ Fukushima-Daiichi release*, **authored by Joffrey Dumont Le Brazidec, Marc Bocquet, Olivier Saunier, and Yelva Roustan**

*Summary*: This paper proposes the use of a reversible-jump Markov chain Monte Carlo sampling algorithm for use in Bayesian inverse problems for source reconstruction. Among the benefits of the method is the ability to capture temporal behavior of the source release in fine detail. The proposed method is applied to source reconstruction for the Fukushima-Daiichi release and results are compared to those from previous studies.

In general, the paper is clearly written. Please see below for a short list of items to address.

1. On line 176, the authors note that "This average wind speed value has not been extensively researched and is only used to estimate the potential of the method." The statement is appreciated. However, is it possible for the authors to comment on any potential sensitivity of the method to values of the wind speed?

2. On lines 275-276, can the authors quantify the terms "good match" and "similar in magnitude"? It will be an aid to the reader when referring to and interpreting the associated figures.

3. Some of the labeling in the figures needs to be fixed:

   (a) Figure 3: the powers are missing (or cut off) for some of the labels on the vertical axis

   (b) Figure 4:

      i. some of the labels on axes are cut off (or incorrect), for example "0." instead of "0.8" on the vertical axis of the first subplot

      ii. the labels indicating magnitude (i.e., the $10, 10^{-1}, 10^1$) are misplaced

      iii. there are several "n"s around the sides/top/bottom of the subplots and it is not clear what they are labeling

   (c) Figure 5: same as for Figure 4 above. In addition, the legend in the first subplot needs to be corrected (they have symbols, not words)

   (d) Figure 6: same as for Figure 5 above.

4. Some typos or suggested rewording:

   (a) end of line 80: The phrase "Here, it is here" should read "Here, it is"

   (b) beginning of line 135: The phrase "intervals are not fixed neither regularly space" could be reworded as "intervals are neither fixed nor regularly spaced"

   (c) end of line 345: the phrase "sample the both source" should read "sample both the source"

---

## Author Comment (AC1)

**Discussion: Bayesian transdimensional inverse reconstruction of the $^{137}$Cs Fukushima-Daiichi release**

Joffrey Dumont Le Brazidec[1,2], Marc Bocquet[2], Olivier Saunier[1], and Yelva Roustan[2]

[1]IRSN, PSE-SANTE, SESUC, BMCA, Fontenay-aux-Roses, France
[2]CEREA, Joint laboratory École des Ponts ParisTech and EDF R&D, Université Paris-Est, Marne-la-Vallée, France

**Correspondence:** Joffrey Dumont Le Brazidec (joffrey.dumont@enpc.fr)

**Report 1**

We would like to thank the anonymous Referee 1 for her/his constructive comments and revisions, which allowed us to clarify several points in the paper.

*Summary: This paper proposes the use of a reversible-jump Markov chain Monte Carlo sampling algorithm for use in Bayesian inverse problems for source reconstruction. Among the benefits of the method is the ability to capture temporal behavior of the source release in fine detail. The proposed method is applied to source reconstruction for the Fukushima-Daiichi release and results are compared to those from previous studies. In general, the paper is clearly written. Please see below for a short list of items to address.*

   – *1. On line 176, the authors note that "This average wind speed value has not been extensively researched and is only used to estimate the potential of the method." The statement is appreciated. However, is it possible for the authors to comment on any potential sensitivity of the method to values of the wind speed?*

   Only one reference wind speed value was tested. To evaluate the potential sensitivity of the method to a change in the reference wind speed, more than ten days of calculation would have been necessary. The lack of improvement of the results using spatio-temporal clustering (or even a slight deterioration) as well as the cumbersome nature of the method led us not to pursue this direction. We have chosen to leave this exploratory work, but this part being much less accomplished than the core of the paper, and not bringing conclusive results, we will follow the recommendation of the reviewers or the editor and delete it if necessary.

   – *2. On lines 275-276, can the authors quantify the terms "good match" and "similar in magnitude"? It will be an aid to the reader when referring to and interpreting the associated figures.*

   The following paragraph has been added to clarify what we mean by «good match»:

   *«Firstly, we observe that a good match is obtained between the median release and the source term of Saunier et al.*

*(2013) or Terada et al. (2020) (on the constrained time intervals), as the differences between releases under $10^9\,Bq.s^{-1}$ can be neglected. Indeed, between the $11^{st}$ and the $19^{th}$, the three source terms have similar peak magnitudes (close to $10^{11}\,Bq.s^{-1}$) and these peaks are observed at close time intervals. Between the $19^{th}$ and the $21^{st}$, a notable difference can be observed: the RJ-MCMC source term predicts very high one-hour peaks while the Terada et al. 2020 source term distributes the release in a globally constant way over the whole period.»*

What matters are the release rates above $10^9$ Bq.s$^{-1}$ observed and those retrieved with the RJ-MCMC tend to be consistent with those reconstructed by Saunier et al. (2013) or Terada et al. (2020).

– *3. Some of the labeling in the figures needs to be fixed: (a) Figure 3: the powers are missing (or cut off) for some of the labels on the vertical axis (b) Figure 4: i. some of the labels on axes are cut off (or incorrect), for example "0." instead of "0.8" on the vertical axis of the first subplot ii. the labels indicating magnitude (i.e., the 10, 10-1, 101) are misplaced iii. there are several "n"s around the sides/top/bottom of the subplots and it is not clear what they are labeling (c) Figure 5: same as for Figure 4 above. In addition, the legend in the first subplot needs to be corrected (they have symbols, not words) (d) Figure 6: same as for Figure 5 above.*

The figures were damaged during the construction of the pdf for an unknown reason.

– *4. Some typos or suggested rewording: (a) end of line 80: The phrase "Here, it is here" should read "Here, it is"*

This has been corrected. Thanks.

– *(b) beginning of line 135: The phrase "intervals are not fixed neither regularly space" could be reworded as "intervals are neither fixed nor regularly spaced"*

This has been corrected. Thanks.

– *(c) end of line 345: the phrase "sample the both source" should read "sample both the source"*

This has been corrected. Thanks.

Thank you very much for all these suggestions.

**Report 2**

We would like to thank the anonymous Referee 2 for her/his technical comments and advices on improving the manuscript.

*Summary: In this work, the authors investigated a transdimensional sampling method to reconstruct highly fluctuating radionuclide atmospheric sources and applied it to assess the 137Cs Fukushima-Daiichi release. The authors apply a reversible-jump Markov chain Monte Carlo sampling algorithm for use in Bayesian inverse problems for source reconstruction. The authors tried various methods in hopes of gaining accuracy and reducing uncertainty in the estimates, such as the inclusion of two observational sources of information (air concentration observations and deposition measurements). The authors found that the total released reconstructed activity is estimated to be between 10 and 20 PBq, which matches previous literature, although this estimate increases when considering the deposition measurements. While the authors do a good job in explaining the methods and the results in the paper, I think there are small points that need to be addressed, mostly grammatical or in the figures. Main comments:*

- *This is just my opinion, but I believe that parts of the introduction that describe the Bayesian inverse modelling approach (e.g. Section 1.2) can be moved to the methods section, and parts of Section 2 describing previous literature on the topic (e.g. Section 2.1) can be moved to the introduction.*

  We prefer to separate the Bayesian inverse modelling approach (1.2) from section 3 for two reasons:

  - in section 1.2 nothing new is introduced (we have already presented this framework in our two previous papers)
  - we need this presentation of the inverse modelling approach to clearly present the transdimensional analysis of this study which should be in the introduction.

  However, following your suggestion, section 2.1 has been moved to the first part of the introduction. It does indeed seem more logical in this position.

- *In general, the metric used to describe the goodness of fit (FAC scores) is hardly described in the text with very little background information. This makes it hard for a reader to judge the accuracy of the results. The authors can do a better job at providing some of this information for clarity. What decides if a given FAC score is considered 'good' or not?*

  Thanks to the reviewer for this very fair comment. Two things have been added in the manuscript to take this into account:

  - a clear definition of the FAC score has been given;
  - the FAC scores of the source terms from Saunier et al. (2013) and Terada et al. (2020) have been given, for comparison. In addition, the discussion around FAC scores has been expanded extensively.

*- The figures seem to have some mistakes or an issue with the processing. Numbers and labels drop off for many of the - figures, please fix this. Also, the main text does not refer to some figures, and in some locations the reference is done with a capital letter (e.g. Figure 1) and in other locations not (e.g. figure 1). Please choose one way and keep consistent.*

The references are now kept consistent over the text. Thanks. The figures were indeed damaged during the construction of the pdf for an unknown reason.

– *Figures 3-5: Is the uncertainty derived as a multiple of the standard deviation? Why not use the information from the posterior samples to apply the uncertainty, can the authors comment?*

We are not sure to understand this comment, as this is what we are doing:

  – on Figure 3 : the light red area corresponds the area between [median – standard deviation, median + standard deviation] of the posterior samples. This is the area giving information about the uncertainty. So we are using the information from the posterior samples to « apply » the uncertainty.

  – on Figure 4d: same thing;

  – on Figure 4a, b, c: these histograms are constructed based on the posterior samples.

*Minor comments:*

  – *Line 7: Is MCMC defined before using here?*

  It was not, we modified the sentence. Thanks.

  – *Line 10: Is 137Cs defined before using here?*

  We have corrected the sentence.

  – *Line 80: Take out the word 'Here'*

  This has been corrected. Thanks.

  – *Lines 113-120: Various assumptions are mentioned here, but it is not clear if the impact of these assumptions on the results is discussed later in the text. It would be great if the authors can provide some comments on how these assumptions can have an influence on the results. These different assumptions about the origin or physical parameterisation of the model are not discussed further in the text.*

  We comment here on the different assumptions:

    – the release occurred mainly during the two or three weeks following 11 March. This follows from the extensive literature on the Fukushima-Daiichi case and we have no reason to discuss it;

    – the longitude/latitude origin of the release is a safe assumption in the case of Fukushi-Daiichi;

- the different assumptions on the physical parameterisation of the model, and the choice of the meteorological fields, which can be found in Table 1. The choices of the meteorological fields and the deposit scheme might have an important influence on the results. We actually published an article in ACP two years ago on, in particular, a method for better evaluating physical parameters using observations:

  https://acp.copernicus.org/articles/21/13247/2021/acp-21-13247-2021.html

  However, it would be complicated to reproduce this method with the reversible jump MCMC in the case of Fukushima-Daiichi (due to the long calculation time). Finally, these parameterisations were chosen following the recommendations of previous studies (Quérel et al., 2021) for the scavenging scheme and (Saunier et al., 2013) for the Kh or the dry deposition rate.

- With regard to the assumption on the height of release, there should be no or only a minor influence because the observations used for the inversion are located far from the source.

– *Line 175: A main assumption is that the average wind speed is representative of the temporally varying wind speed used in the simulations. Is this an average of the whole period? What effect does this have on the error/uncertainty of the results, compared to using a time-variable input vector for wind speed that is derived directly from observations? Again, some comments from the authors would be appreciated.*

The average wind speed used is not the average of the wind value over the whole period. This value of 12 km.h$^{-1}$ is representative in order of magnitude of the average wind but we did not try to refine this value. This value was only considered to evaluate the potential of the method. The lack of improvement in the results using this spatio-temporal clustering (or even a slight deterioration) as well as the cumbersome nature of the method led us not to continue in this direction, i.e. not to refine this value and not to try to evaluate it with a time-varying wind. We therefore only kept a spatial clustering (which gave conclusive results). Finally, we have chosen to leave this exploratory work, but this part being much less accomplished than the core of the paper, and not bringing conclusive results, we will follow the recommendation of the reviewers or the editor and delete it if necessary.

– *Line 204: take out the word 'for'*

This has been corrected. Thanks.

– *Line 271: "Figure 3 shows . . . ? " There is a grammatical mistake here, please fix.*

This has been corrected. Thanks.

– *Line 275-285: The authors describe a 'good match' and 'similar in magnitude', but how can the reader quantify if this is true? Are the FAC scores supposed to represent this? If that is the case, the current information on the FAC score is not sufficient for the reader to make these conclusions. More information on the FAC score is needed.*

The following paragraph has been added to clarify what we mean by «good match»:

«Firstly, we observe that a good match is obtained between the median release and the source term of Saunier et al.

(2013) or Terada et al. (2020) (on the constrained time intervals), as the differences between releases under $10^9 \, \mathrm{Bq.s}^{-1}$ can be neglected. Indeed, between the $11^{\mathrm{st}}$ and the $19^{\mathrm{th}}$, the three source terms have similar peak magnitudes (close to $10^{11} \, \mathrm{Bq.s}^{-1}$) and these peaks are observed at close time intervals. Between the $19^{\mathrm{th}}$ and the $21^{\mathrm{st}}$, a notable difference can be observed: the RJ-MCMC source term predicts very high one-hour peaks while the Terada et al. (2020) source term distributes the release in a globally constant way over the whole period.»

This expression of "good match" is visual so the reader can quantify it directly from the various graphs. The FAC score is only meant to assess the overall quality of the source term but not to compare it to other source terms. Based on your feedback, we provide more information about the source term in the text.

- *Line 345: "Both the" instead of "the both"*

  This has been corrected. Thanks.

- *Line 346-347: The authors state that "an adaptive grid allows to reconstruct the source term with higher accuracy and to reduce the corresponding uncertainties". Was the higher accuracy and reduced uncertainty actually achieved? Where does the reader get this information from?*

  This sentence in the conclusions section is used to describe what the RJ-MCMC can do, theoretically. However, it is true that this has not been proven in the manuscript. We have therefore changed the sentence to

  «In the case of complex releases, an adaptive grid allows to reconstruct the source term with high accuracy and discretisation consistent with the available observations.»

  The work was nevertheless done: Olivier Saunier (one of the co-author) performed a deterministic inversion on the air concentrations and obtained an FAC2 score of 0.27 on the air concentrations (<0.31 with the RJ-MCMC). This calculation is based on the approach used in Winiarek et al. (2012) This work is not described in the manuscript because it has not been published. Moreover, the main interest of the Reversible Jump MCMC does not lie in obtaining a maximised deterministic score, but in a more accurate representation (discretisation) of the source term.

Thank you very much for all these suggestions.

**Comments from the Executive Editor**

We would like to thank the Executive Editor for her two well justified suggestions.

- *Dear authors,*

  *as a model for geoscientific model development, one important part of the journal is the publication of the algorithm implementations and models used. Please refer to*

  *https://www.geoscientific-model-development.net/policies/code_and_data_policy.html for our code and data policy. In your article the Code availability section is missing completely. Please provide your algorithm implemenations and the models used (i.e., "the Eulerian model ldX") to the readers. They should be made available within a publicly accessible permanent archive (e.g. Zenodo).*

  *Best regards,*

  *Astrid Kerkweg (GMD Executive Editor)*

  The algorithm IBISA (Inverse Bayesian Inference for Source Assessment) which describes in particular the Reversible-Jump Markov Chain Monte Carlo algorithm used in this article is available via the following zenodo repository:

  https://zenodo.org/record/7318543 or 10.5281/zenodo.7318543

  It is also available with descriptions on github:

  https://github.com/JoffreyDumontLeBrazidec/ibisa/releases/tag/v1.0.0

  This repository also includes a data directory containing

    - an observation operator (result of the ldX simulations),

    - a vector of observations,

  which are provided to run the algorithm.

  Providing access to the ldX code is more intricate as the code is the property of the IRSN (Institut de radioprotection et de sûreté nucléaire). However, the ldX model is only used to compute the observation operator matrix (which is available in the zenodo repository). Moreover, the ldX model is derived from the open-source Air Quality Modeling System Polyphemus model (see http://cerea.enpc.fr/polyphemus/). The Polyphemus codes can be found on gitlab at https://gitlab.com/polyphemus/. Note that we use ldX and Polyphemus/Polair3D interchangeably for this collaborative research effort on radionuclides transport and fate.

- *Please ensure that the colour schemes used in your maps and charts allow readers with colour vision deficiencies to correctly interpret your findings. Please check your figure 3 using the Coblis – Color Blindness Simulator (https://www.color-blindness.com/coblis-color-blindness-simulator/) and revise the colour schemes accordingly.*

  Figure 3 seemed to be correct but we nevertheless replaced it with a clearer colour map scheme. However, the two colours in the last figure could not be distinguished for Monochrome/Achromatopsia, so the colours were changed.

Thank you very much for these suggestions.

**References**

Quérel, A., Quélo, D., Roustan, Y., and Mathieu, A.: Sensitivity study to select the wet deposition scheme in an operational atmospheric transport model, Journal of Environmental Radioactivity, 237, 106 712, https://doi.org/10.1016/j.jenvrad.2021.106712, publisher: Elsevier, 2021.

Saunier, O., Mathieu, A., Didier, D., Tombette, M., Quélo, D., Winiarek, V., and Bocquet, M.: An inverse modeling method to assess the source term of the Fukushima Nuclear Power Plant accident using gamma dose rate observations, Atmos. Chem. Phys., 13, 11 403–11 421, https://doi.org/10.5194/acp-13-11403-2013, 2013.

Terada, H., Nagai, H., Tsuduki, K., Furuno, A., Kadowaki, M., and Kakefuda, T.: Refinement of source term and atmospheric dispersion simulations of radionuclides during the Fukushima Daiichi Nuclear Power Station accident, J. Environ. Radioact., 213, 106 104, https://doi.org/10.1016/j.jenvrad.2019.106104, 2020.

Winiarek, V., Bocquet, M., Saunier, O., and Mathieu, A.: Estimation of errors in the inverse modeling of accidental release of atmospheric pollutant: Application to the reconstruction of the cesium-137 and iodine-131 source terms from the Fukushima Daiichi power plant, J. Geophys. Res. (Atmospheres), 117, D05 122, https://doi.org/10.1029/2011JD016932, 2012.

---

## Referee Report (RR1)

**Review of gmd-2022-168:** *Bayesian transdimensional inverse reconstruction of the $^{137}Cs$ Fukushima-Daiichi release*, **authored by Joffrey Dumont Le Brazidec, Marc Bocquet, Olivier Saunier, and Yelva Roustan**

*Summary*: This paper proposes the use of a reversible-jump Markov chain Monte Carlo sampling algorithm for use in Bayesian inverse problems for source reconstruction. Among the benefits of the method is the ability to capture temporal behavior of the source release in fine detail. The proposed method is applied to source reconstruction for the Fukushima-Daiichi release and results are compared to those from previous studies.

*Comments for authors*: Thank you very much for addressing the items from the first round of reviews. I am satisfied with the changes made and the responses from the authors. I have no further requests for the authors to address, except for a typo that needs to be corrected:

line 276 of the revised manuscript: "are observed neighbouring time intervals" should read "are observed *at* neighbouring time intervals"